# Wearable Capacitive Pressure Sensor for Contact and Non-Contact Sensing and Pulse Waveform Monitoring

**DOI:** 10.3390/molecules27206872

**Published:** 2022-10-13

**Authors:** Azmal Huda Chowdhury, Borzooye Jafarizadeh, Nezih Pala, Chunlei Wang

**Affiliations:** 1Department of Mechanical and Materials Engineering, Florida International University, Miami, FL 33174, USA; 2Department of Electrical and Computer Engineering, Florida International University, Miami, FL 33174, USA; 3Center for Study of Matter at Extreme Conditions, Florida International University, Miami, FL 33174, USA

**Keywords:** capacitive sensor, porous PDMS, MWCNT, wearable pressure sensor, heart rate monitor, proximity sensing

## Abstract

Sensitive and flexible pressure sensors have invoked considerable interest for a broad range of applications in tactile sensing, physiological sensing, and flexible electronics. The barrier between high sensitivity and low fabrication cost needs to be addressed to commercialize such flexible pressure sensors. A low-cost sacrificial template-assisted method for the capacitive sensor has been reported herein, utilizing a porous Polydimethylsiloxane (PDMS) polymer and a multiwalled carbon nanotube (MWCNT) composite-based dielectric layer. The sensor shows high sensitivity of 2.42 kPa^−1^ along with a low limit of detection of 1.46 Pa. The high sensitivity originates from adding MWCNT to PDMS, increasing the composite polymer’s dielectric constant. Besides this, the pressure sensor shows excellent stability at a cyclic loading of 9000 cycles, proving its reliability for long-lasting application in tactile and physiological sensing. The high sensitivity of the sensor is suitable for the detection of small deformations such as pulse waveforms as well as tactile pressure sensing. In addition, the paper demonstrates a simultaneous contact and non-contact sensing capability suitable for dual sensing (pressure and proximity) with a single data readout system. The dual-mode sensing capability may open opportunities for realizing compact systems in robotics, gesture control, contactless applications, and many more. The practicality of the sensor was shown in applications such as tactile sensing, Morse code generator, proximity sensing, and pulse wave sensing.

## 1. Introduction

Flexible wearable pressure sensors have garnered tremendous attention in recent years due to their benefits in many applications, including healthcare monitoring [1], electronic skin [2], tactile sensing [3], electronic textiles [4], and soft robotics [5]. Numerous flexible pressure sensors have been developed so far, capable of low-pressure (<10 kPa) and high-pressure (>10 kPa) detection applications, such as monitoring pulse waveforms from wrist arteries (0–10 kPa) and tactile pressure sensing (10–100 kPa) [6,7]. Significant focus has been given to flexible pressure sensors for cardiovascular monitoring from arterial wrist sites in recent years. Cardiovascular disease (CVD) is one of the most prominent causes of the global health crisis. CVD-borne diseases such as hypertension are responsible for premature death worldwide, necessitating continuous monitoring of cardiovascular activities [8]. Cardiovascular biomarkers such as heart rate and pulse waveforms can be monitored by smart wearable devices such as a smartwatch. Specifically, pulse waveforms contain vital information about cardiovascular health by analyzing the shape of the peaks of the pulse waveforms. However, smart wearable devices use Photoplethysmography (PPG)-based optical techniques, which have limitations in resolving the pulse waveforms accurately, resulting in the shortcomings of the pulse waveform analysis [9]. On the other hand, recent advances in flexible pressure sensors allow resolving the pulse waveforms to their characteristic peaks, making them an attractive solution for monitoring pulse waveforms. These pressure sensors are categorized into capacitive, piezoresistive, piezoelectric, and triboelectric sensors [10,11,12]. Among these different sensors, capacitive pressure sensors (CPSs) have a simple architecture that requires a dielectric layer sandwiched between two parallel plate electrodes [13]. Besides this, CPSs require a low operating voltage and have immunity against temperature with a simple data readout system [14,15]. In addition to pressure sensing, there is an increasing need to leverage different sensing mechanisms to mimic human sensory organs. Adding several sensors demands individual sensing output systems, making achieving such a goal complex and challenging [16]. CPSs can perceive proximity using the fringe field [17]. However, proximity detection significantly reduces pressure sensitivity, which limits the application of the sensor for both high-performance pressure sensors and proximity sensors [18]. Therefore, there is a need to realize high-performance pressure sensors with both high sensitivity and proximity sensing capabilities.

The sensitivity of pressure sensors is an essential performance indicator. The sensor materials and fabrication must be carefully selected and optimized to realize pressure sensors with high sensitivity while keeping the cost acceptable. However, achieving superior performance requires meticulous fabrication techniques, such as photolithography, which can be costly and lengthy, involving multiple steps [19,20]. Therefore, developing a simple and low-cost fabrication method is essential for commercial applicability. A CPS based on a parallel plate mechanism with a solid polymer-based dielectric layer has demonstrated low sensitivity in both low- and high-pressure applications [10,14,21]. There have been many efforts toward improving pressure sensitivity, especially in the low-pressure regime, to detect minute pressure, such as the pulse waveforms from the wrist artery. These efforts are concentrated on improving the dielectric/electrode layer’s structural properties and the polymer layer’s dielectric properties. It has been shown previously that the presence of microstructures on the sensing layer improves the pressure sensitivity on both capacitive and piezoresistive pressure sensors [22,23]. On the one hand, to reduce the stiffness of the dielectric layer/electrodes, different microstructuring approaches were adopted, such as using photolithography, nature-based template-assisted mold fabrication technique, and 3D printing to realize micro pyramid structure [24], micro dome [22], and micropillar structures [25]. However, materials with such delicate surface microstructures typically possess low reproducibility and reliability under heavy loads, which degrades the sensing performance [26]. Besides, the pressure sensors with a microstructured sensing layer have high sensitivity at low pressure, but the sensitivity drops significantly at high-pressure [22,27]. For instance, Zhuo et al. reported a 3D-printed template-assisted micropatterning technique to realize a capacitive pressure sensor with a microstructured sensing layer [27]. Although the sensor could achieve a high sensitivity of 1.62 kPa^−1^ below 200 Pa, the sensitivity drops significantly to 0.05 kPa^−1^ between 1 and 4 kPa. Moreover, creating such microstructures on the dielectric layer/electrode requires complicated, multi-step, and expensive fabrication processes [28]. On the other hand, researchers tried different natural template-assisted mold fabrication techniques to realize micropatterned sensing layers for the CPSs [29]. Although such template-assisted methods are low-cost alternatives to micropatterned sensing layer fabrication techniques, they suffer from reliability and reproducibility issues, since ensuring the uniformities of the templates for batch production is challenging. Therefore, the issue of a reliable and low-cost approach to high-sensitivity pressure sensors needs to be addressed. An alternative approach to improve the performance of CPSs with low-cost techniques is by reducing the stiffness of the dielectric layer by introducing pores inside the dielectric layer [30]. Several methods have been implemented over recent years to realize polymer layers with porous/foamy structures, including foaming [14], emulsion [17,28], dip coating [31,32], and sacrificial template [10,33] methods. Sacrificial template-assisted porous dielectric layer fabrication is a suitable and low-cost approach to reducing the stiffness of the dielectric layer without any complicated micropatterning techniques. The stiffness reduction technique is achieved by introducing pores inside the polymer dielectric layer by adding different sacrificial templates such as sugar [10,30,33], salt, and PS beads [3]. For instance, Kang et al. reported a bio-inspired porous dielectric layer-based CPS using polystyrene beads as the sacrificial template [3]. The sensor achieved the highest-pressure sensitivity of 0.63 kPa^−1^ and an LoD of 2.42 Pa.

Nevertheless, the pressure sensitivities of porous dielectric layer-based CPSs are typically lower compared to other more structurally ordered micropattern-based CPSs [34]. The sensing performance can be further improved by functionalizing the porous/foamy dielectric layer with high dielectric constant dopants such as CaCu_3_Ti_4_O_12_ (CCTO) [32,35] or by adding highly conductive nanofiller materials such as carbon nanotubes [36], carbon black [37], graphene nano platelets [31], etc. According to previous reports, adding high-conductivity nanofiller materials could improve sensitivity more effectively than adding high-dielectric permittivity dopants [38]. In particular, the dielectric constant of the polymer material can be significantly improved by adding a conductive nanofiller below the percolation threshold that improves the pressure sensitivity of the CPS. Therefore, utilizing conductive filler materials with a porous dielectric layer can be a facile route and low-cost technique to improve the pressure-sensing performance of CPSs. Additionally, the sensitivity improvement of such low-cost pressure sensors can successfully detect low-pressure stimuli, such as pulse waveforms from wrist arteries.

Herein, we introduce for the first time a low-cost sacrificial template-assisted method to realize porous PDMS-MWCNTs (polydimethylsiloxane-multiwalled carbon nanotubes) dielectric layers to improve the pressure sensitivity of the CPS. The porous dielectric layer shows excellent mechanical compressibility and flexibility suitable for application in wearable pressure sensors. The sensor also shows long-term cyclability and a very low-pressure detection limit. In addition, the CPS shows excellent proximity detection capability in a non-contact mode which is suitable for simultaneous pressure and proximity detection using a single data readout system. In order to demonstrate the sensor’s practicality, we applied our flexible pressure sensor for tactile pressure sensing, generating Morse code, proximity sensing, and acquiring pulse waveforms from the arterial wrist site.

## 2. Results and Discussion

### 2.1. Fabrication of Porous Dielectric Layer

Figure 1 shows the schematic illustration of the MWCNT-PDMS-based dielectric layer fabrication process. Sucrose particles were used as the sacrificial template for creating porosity inside the polymer layer. The fabrication process has three steps. A PDMS solution was prepared and then mixed with MWCNTs dispersed in acetone. After mixing the solution, sucrose particles were added to the mixture solution, as shown in Figure 1a(i). The entire solution was cast on a glass mold to get the desired shape of the dielectric layer, as shown in Figure 1a(ii). After curing the solution, the sucrose particles were dissolved in water under magnetic stirring, leaving a porous composite layer behind (Figure 1a(iii,iv)). Figure 1b shows the loading-unloading scenario of the sensor. Due to pores inside the dielectric layer, the air trapped inside leaving the pores could generate a compressed dielectric layer when external loading is applied.

### 2.2. Materials Characterization

Figure 2a shows the schematic of the fabricated sensor. The sensor includes an MWCNT-PDMS-based dielectric layer sandwiched between two conductive textile-based electrodes. The sensor was packaged inside Kapton tape for practical use. Figure 2b shows SEM images of the porous MWCNT-PDMS dielectric layer with different CNT concentrations and the conductive textile electrode. According to the SEM images, the pores are randomly distributed throughout the entire volume of the porous dielectric layer. The pore size and volume can be controlled by changing the size and the number of sucrose particles during the fabrication. However, according to a previous report, excessive pores may destroy the mechanical behavior of the pressure sensor and reduce its performance [39]. The sacrificial template’s volume significantly affects the pore volume of the porous dielectric layer. Besides this, adding acetone to the PDMS increases the interconnected pore volume since the acetone evaporates during the curing process. The pore size from the SEM image was calculated to be 455.22 ± 168.92 μm. The SEM images do not show residual sucrose particles inside the polymer matrix. The SEM image of the conductive textile shows the presence of high roughness, which may add to the high sensitivity of the CPS (Figure 2b(iv)).

FTIR analysis was carried out to characterize the interaction between the PDMS and MWCNTs-based dielectric layer, as shown in Figure 2c. The peaks around the 785–815 cm^−1^ and 835–855 cm^−1^ regions represent Si(CH_3_) rocking bands and Si-C bonds, respectively [40]. A wide stretching of peak ranges from 1000 cm^−1^ to 1100 cm^−1^ representing a symmetrical Si-O-Si stretching bond. The peaks at 1410 cm^−1^ and 1258 cm^−1^ are -CH_3_ symmetric and asymmetric deformations, respectively. The peaks around 2900 cm^−1^ and 2960 cm^−1^ correspond to the CH_3_ symmetric and asymmetric stretching [41]. As the concentration of the MWCNT increases, the absorbance peak ratio around 900 cm^−1^–930 cm^−1^ between samples with increasing MWCNT content increases, which agrees with the literature [42]. Figure 2d shows photographs of pure PDMS, and MWCNT-PDMS composites, and the fabricated sensor. The composite polymer appears darker than the pure counterpart due to the CNT inside the polymer matrix. The pictures also demonstrate that the presence of the pores inside the polymer matrix results in excellent flexibility and compressibility of the dielectric layer during bending.

### 2.3. Electromechanical Characterization

The pressure-sensing mechanism of the CPS can be explained with the help of the parallel plate capacitor mechanism [38]. According to the parallel plate mechanism, the capacitance depends on the distance (*d*) between the parallel plates, the electrode area (*A*), and the dielectric permittivity (*ε*). The initial capacitance before applying any pressure on the capacitive sensor is:
(1)$$C_{0}=\frac{\varepsilon_{0}\varepsilon_{r}A}{d}.$$
where *C*_0_ is the base capacitance, *ε*_0_ is the dielectric permittivity of vacuum, *ε_r_* is the dielectric permittivity of the dielectric layer, *A* is the electrode area, and d is the separation distance between the electrodes. Once pressure is applied, the new capacitance under pressure becomes:(2)$$C_{0}\;+∆C=\frac{\left(\varepsilon +\Delta \varepsilon \right)A}{\left(d-\Delta d\right)}.$$
where Δ*C* is the change in capacitance, Δ*ε* is the change in dielectric permittivity, and Δ*d* is the change in distance between the electrodes. Although the capacitance depends on the area of the electrodes, during compression, the electrodes do not go through a significant lateral area change. Therefore, the sensitivity of porous MWCNT-PDMS-based sensors depends mainly on the change in separation distance between the electrodes and the change in dielectric permittivity.

In order to improve the pressure response of the CPSs, the dielectric layer should deform easily under slight pressure to induce enough change in capacitance. Moreover, the dielectric permittivity should be high to achieve a high signal level from the sensor. The porous MWCNT-PDMS-based CPSs could achieve both objectives by reducing the Young’s modulus of the dielectric layer and by improving the dielectric permittivity of the sensor. Under pressure, the separation distance ‘*d*’ decreases with ease due to the pores inside the dielectric layer. The overall porosity of the dielectric layer affects the performance of the CPS significantly. Research shows that the pore size and overall pore volume directly influence the sensitivity of the CPS. As the pore size increases, the pressure sensitivity of the CPS improves due to the better compressibility of the dielectric layer [28,30]. However, very large increments in the pore size may negatively impact the polymer’s structural property, reducing the mechanical stability of the CPS [39]. Such inferior mechanical stability may cause the low recoverability of the CPS.

Here, controlling the pore volume was adopted instead of controlling the pore size to reduce the stiffness of the dielectric layer, as commercial sucrose particles were used as the sacrificial layer without any further treatment. The total pore volume comes from the sacrificial sucrose particles and volatile acetone during the curing process. During the study, the pore volume was kept constant throughout the samples by keeping a fixed ratio between the PDMS, sucrose, and acetone, and a pore volume of ≈82% was obtained. For pure porous PDMS-based CPSs, with increasing pressure, the dielectric permittivity increases due to the gradual replacement of air (*ε*_0_ = 1.0) with a higher dielectric constant PDMS (*ε_r_* ≈ 3.0). For MWCNT-PDMS-based CPSs, the change in dielectric permittivity is more significant than for pure PDMS-based CPSs. According to a previous study, with additions of MWCNTs beyond 0.9 wt.%, the dielectric permittivity of solid PDMS-MWCNTs composite increases significantly (*ε_r_* ≈ 100) at 1 kHz frequency [43]. Without the addition of MWCNTs, the dielectric permittivity of the pure porous PDMS is 3.0. As the concentration increases, the permittivity starts to increase as expected. After adding 0.4% of MWCNTs, the permittivity increases to 4.109. With a subsequent increment of the MWCNTs, the permittivity increases by 4.988, 5.05, 5.15, and 5.2415 for 0.8%, 1.2%, 1.6%, and 2% of MWCNTs, respectively. This increment in dielectric permittivity comes from the combined air/PDMS permittivity under no pressure. However, as the pressure increases, the dielectric permittivity also changes, leading to a more considerable capacitance change that improves the pressure sensitivity of the sensor. However, at higher loading of the MWCNTs, the dielectric permittivity increases significantly, leading to high base capacitance and stiffness. Therefore, the pressure sensitivity decreases significantly.

Figure 3 shows the schematic of the testing setup for carrying out the electromechanical tests of the CPSs. The setup contains a mechanical test stand that applies force on top of the CPSs, and the corresponding capacitance signal is measured in an LCR meter. The data from the LCR meter are transmitted to a user interface via LabVIEW for visualization and further processing. Figure 4 shows the electromechanical performances of the CPS based on porous MWCNT-PDMS dielectric layers. Figure 4a and Appendix A show the sensor’s base capacitance (*C*_0_) as a function of the MWCNT concentration. For statistical analysis, multiple sensors were measured for base capacitance calculation with the same MWCNT concentration at a frequency of 1 kHz and a voltage of 1 V. Then, the base capacitance (*C*_0_) was plotted as a function of the MWCNT concentration as the *x*-axis. As the filler concentration increases, the base capacitance of the sensor increases due to the high dielectric constant of the material. The enhancement in dielectric constant happens due to dipole formation and micro capacitor network formation [6,43]. Since the capacitance is directly proportional to the dielectric permittivity of the sensor, the base capacitance increases. Figure 4b shows the pressure sensing performance of the sensors with different filler concentrations. The pressure sensing performance was characterized by plotting the relative change in capacitance vs. the external pressure. The pressure sensitivity for a CPS is defined as:(3)$$S=\frac{\Delta C/C_{0}}{\Delta P},$$
where ∆*C*/*C*_0_ is the relative capacitance change, and Δ*P* is the applied pressure [7]. The unit of pressure sensitivity is defined as kPa^−1^, while the change in capacitance is normalized to the base capacitance to make it dimensionless. This approach is well adopted since the performance of different pressure sensors (i.e., capacitive, piezoresistive) can be compared using this formula. The figure shows that the pressure sensitivity increases with the addition of MWCNT with the PDMS. The highest sensitivity was obtained from the sensor having 1.6% MWCNT with a high sensitivity of 2.41 kPa^−1^ under 0.5 kPa, and 0.11 kPa^−1^ beyond 0.5 kPa.

In comparison, the pressure sensor with no MWCNT achieved a sensitivity of 0.31 kPa^−1^ below 0.5 kPa, and 0.07 kPa^−1^ beyond 0.5 kPa. The initial high sensitivity can be explained by considering the initial high volume of the pore size inside the dielectric layer. The presence of the pores significantly reduces the PMDS’s elastic modulus, improving the polymer’s pressure-sensitive response. Therefore, a small force can generate enough deformation in the porous dielectric layer in the low-pressure range. However, such enhancement in the pressure response does not significantly improve the pressure sensitivity due to the low dielectric permittivity of the PDMS. Adding MWCNTs improves the polymer’s dielectric permittivity significantly, thereby improving the pressure sensitivity. As a result, a high capacitance signal can be achieved. As the pores collapse quickly, leaving a high dielectric constant elastomer upon compression, the pressure sensitivity decreases. The pressure sensitivity increases with the addition of MWCNT up to 1.6%, after which the pressure sensitivity decreases, as shown in Figure 4b and Appendix A. The reduction in pressure sensitivity can be discussed with the help of Figure 4a and Appendix A. With the addition of MWCNT to the PDMS, the base capacitance increases. At the same time, the capacitance signal under compression (∆C) also increases. However, since the pressure sensitivity is calculated by relative capacitance, the ∆*C*/*C*_0_ increases first. It then decreases because the ∆C part cannot keep up significantly enough with the base capacitance *C*_0_ to enhance the pressure sensitivity further. Another important reason behind the decrease in the sensitivity at higher concentrations of MWCNTs is the increased stiffness that comes from adding the MWCNTs [44]. As the concentration of MWCNTs increases, both stiffness and dielectric permittivity control the pressure sensitivity. At 1.6% MWCNTs concentration, the sensor may have achieved the highest sensitivity due to the tradeoff between the permittivity and stiffness issue. As the concentration increases, the base capacitance jumps significantly after 3% MWCNT concentration leading to a significant decrease in sensor performance (Appendix A). Due to such a complex relationship, the highest sensitivity was obtained from the sensor with 1.6 (wt.%) of MWCNTs to the PDMS polymer solution (Figure 4c). To further show the capacitance response of the sensors with different MWCNT concentrations, the samples were cycled at 1 kPa. Figure 4d shows the capacitance signal of samples having 0%, 0.8%, and 1.6% concentrations of MWCNT. The results show that the sensor with a 1.6% concentration of MWCNT could achieve the highest ∆*C*/*C*_0_. This result is per Figure 4b, where the plot shows the highest sensitivity comes from the sensor having 1.6% of MWCNT.

As a result, the CPS with 1.6% of MWCNTs was chosen for further electromechanical characterization. Figure 4e shows the loading-unloading profile for the MWCNT-PDMS-based sensor with 1.6 wt.% of MWCNTs under different levels of pressure. The sensor’s response at different pressures shows the sensor can detect different pressure levels with adequate change in ∆*C*/*C*_0_. The sensor was also evaluated to detect ultralow pressure, as shown in Figure 4f. For that, it was sequentially loaded with different masses to apply the minimum available pressure of 1.4 Pa, which is equivalent to a mass of 69 mg (about twice the weight of a grain of rice) on the sensor surface area (2.2 × 2.2 cm^2^). The figure shows that the sensor can effectively detect the sequential loading of 1.4 Pa of pressure with a cumulative sum of 7.24 Pa. Therefore, our sensor’s confirmed limit of detection (LoD) is 1.46 Pa.

To show the reliability of the fabricated CPS, it was cycled 9000 times, as shown in Figure 5a. The plot shows that the initial and final cycles are stable without significant performance degradation. The cyclic profile became stable after the first initial cycles and plotted after the stabilization. The cyclic tests show the pressure sensor’s excellent stability, proving its long-term functionality. Moreover, the pressure sensor was compared with a commercial piezoresistive pressure sensor. The commercial sensor was used to track the input load, and the CPS’s output signal was studied to check if the CPS could follow that input loading signal. Figure 5b shows the sensor performance of both commercial and capacitive sensors.

Both sensors were tested on the same test bed we built to compare the sensing performance between the CPS and a commercial sensor. The CPS can maintain a linear shape during the profile’s rise and fall times in both profiles. However, the CPC signal is lagging compared to the piezoresistive pressure sensor. Figure 5c shows an enlarged profile image to show how much the CPS lags, corresponding to the piezoresistive pressure sensor. During the signal’s rise time, there is a delay of 0.19 s between the capacitive and piezoresistive signals. The delay time increases during the fall time of the sensor. The sensing mechanism of CPSs can explain this. The CPS relies on the parallel plate sensing mechanism. The dielectric layer is compressed as the sensor is compressed, leading to the capacitive response. As the pressure is removed, the dielectric material returns to its initial state. However, the polymer material has a viscoelastic effect that is prominent in the solid polymer dielectric layer. The viscoelasticity is reduced with the presence of the pores inside the dielectric layer as the air has negligent viscoelasticity. As a result, the viscoelasticity is reduced significantly for the porous polymer layer. Due to the viscoelasticity, the dielectric material takes some time to catch up, which could explain why the delay time is increased during fall time.

### 2.4. Application

The sensor was evaluated for different real-time applications to demonstrate its outstanding performance. Due to high-pressure sensitivity and excellent stability, the pressure sensor has potential in different physiological, and tactile sensing applications. What is more, the sensor demonstrated proximity sensing capability for contactless sensing applications such as touch sensors. The high sensitivity is suitable for obtaining weak physiological information such as pulse waveform collection from the wrist artery (Figure 6a). To get the pulse waveforms from the wrist artery, the sensor was attached to the top of the wrist artery conformably using a bandage. The data were collected from a 30-year-old healthy male volunteer under normal conditions. The pulse rate from the wrist artery shows a pulse rate of 85 beats per minute. The complete range of the pulse waveform collection is shown in Appendix A. Pulse waveforms carry a significant amount of information about the cardiovascular system. The continuous monitoring of pulse waveforms can help with the early diagnosis of cardiovascular anomalies such as hypertension, arterial fibrosis, and arrhythmia. The pulse waveforms show three distinct peaks characterized by the systolic peak (P-wave), diastolic peak (D-wave), and percussion response (T-wave) from the peripheral arteries as shown in Figure 6b. These peaks convey important information about the cardiovascular system. The augmentation index (AIx) is the ratio between the P-wave and T-wave and is an accepted measure of arterial stiffness and cardiovascular risk factor [45]. The AIx of the male volunteer was found to be 68% which represents no vascular aging in the subject [46,47].

Tactile sensing is essential for numerous applications, including prosthetics, the diagnosis of Parkinson’s disease, robotic hands, electronic skin, and many more. Figure 6c shows low-frequency tactile pressure monitoring. To monitor the pressure variation during tactile pressing, the sensor was placed on top of a digital weight and tapped at irregular intervals. The figure shows that the sensor has a fast response to the tactile pressure and the variance in the signal corresponds to the light and heavy tapping. Due to the fast tactile response of the sensor, the sensor can be employed for generating Morse code from finger knocking, for the diagnosis of Parkinson’s disease, etc. For instance, Figure 6d shows a Morse code application of the pressure sensor. Using the Morse code technique, the sensor was tapped to represent different English words. Morse code is generated using short marks (dots) and long marks (dashes). The practical application of such a device can be in a hospital setting where paralyzed patients can express their feelings using Morse code.

A unique feature of CPSs is detecting proximity [17]. The capacitive fringe effect-based proximity sensor is a more attractive solution for small proximity detection than ultrasound sensors and cameras due to the large power consumption in ultrasound sensors and the limited depth perception in single focal point cameras [16]. Moreover, the ability to detect pressure and proximity using a single data readout system makes the sensing system simple without requiring multiple sensors and different data readout circuitry for simultaneous pressure and proximity sensing applications. Appendix A shows the proximity sensing mechanism for our CPS. The electrode voltage in a parallel plate capacitor creates an electric field between them. This electric field is concentrated between the parallel plates and extends to some distance away, denoted by the fringe field. The proximity effect is due to the disturbance of the electric field lines (fringe field) when a conductive or dielectric material (e.g., a human finger) comes near a capacitive sensor, which decreases the capacitance [16,48]. When a finger approaches the sensor, part of the fringe field is absorbed, resulting in decreased capacitance [18]. As the hand retreats, the fringe field is restored, thereby increasing the capacitance. The realized CPS shows an excellent proximity sensing capacity that could be useful for hands-free applications (Figure 6e). As depicted in Figure 6**e**, the relative capacitance decreases by 8% when a human finger approaches the CPS. The proximity effect range was determined by using a scale and by moving the hand out from the proximity of CPS. The fabricated sensor has a proximity detection range of 12 inches. It can detect any conductive object within this range, which is very suitable for applications such as touch and gesture sensing, electronic skin, tactile sensors for robots, etc. For instance, tactile sensing robots can improve movement by incorporating the proximity sensor, which can control the speed of the movement to avoid collision [16]. Due to the surge in the COVID-19 situation and to ensure public safety by preventing the spread of the disease via touch, automatic applications such as automatic disinfection dispensers, automatic soap dispensers, automatic water faucets, automatic door handles, and many more applications may benefit from such a proximity sensing technique.

Table 1 compares recent studies with porous dielectric layer-based sensing mechanisms. Our porous PDMS/MWCNT-based pressure sensor achieved higher sensitivity in both low-pressure and high-pressure regimes.

## 3. Materials and Methods

### 3.1. Materials

A PDMS elastomer was purchased from Dow Corning. A SYLGARD™ 184 Silicone Elastomeric kit and MWCNT with an average size (O.D. × L) of 6–13 nm × 2.5–20 μm were purchased from cheaptubes.com. Acetone, and sucrose particles were purchased from Fisher Scientific. Conductive carbon cloth was purchased from MSE supplies. All the chemicals were used as received.

### 3.2. Fabrication of Porous PDMS

The PDMS solution with base and curing agents at a 10:1 ratio (*v*/*v*) was mixed in a pre-cleaned beaker. The MWCNTs were weighted as a percentage of the weight to PDMS, added to acetone, and dispersed mechanically. For this research, five different samples were prepared with the weight percentage of the MWCNTs at 0%, 0.4%, 0.8%, 1.2%, 1.6%, and 2%, respectively. The PDMS mixture and the MWCNT solution in acetone were mixed until a homogeneous mixture solution was obtained. Finally, sucrose particles were added to the mixture solution. The volume ratio between PDMS, acetone, and sucrose particles was kept at 2:1:8 (PDMS: Acetone: Sucrose) for all samples reaching a pore volume of ≈82%. Finally, the mixture solution was poured into a glass mold, and the solution was cured for 5 h at 70 °C. The dimension of the glass mold was 40 mm × 25 mm × 2 mm. After curing the polymer, the sucrose particles were dissolved in water overnight, leaving highly porous polymer films behind.

### 3.3. Sensor Fabrication

The porous PDMS/MWCNT-based pressure sensor was fabricated with a layer-by-layer stacking process. The cured dielectric layer was cut into 1 cm × 1 cm and placed on the conductive carbon cloth electrodes. Another conductive carbon cloth electrode was used to complete the sensor fabrication. For electrical wiring, copper tape was pasted on the edge of the carbon cloth and soldered to an external wire. Finally, the sensor was packaged inside polyimide tape.

### 3.4. Characterization and Measurement

The structure and morphology of the porous polymer were characterized using an SEM (JSM-FS100). FTIR was conducted using JASCO FT/IR-4100 to characterize different concentration composite samples. The electromechanical characterization was conducted using a MARK-10 ES-20 test stand connected to a MARK-10 M5-50 force gauge for precise loading. The test stand can apply normal force on top of the device under test (DUT) and the force value is shown on the force gauge. To apply uniform pressure on the sensor, a thin glass slide of 2.2 cm × 2.2 cm was used between the force gauge and the sensor. The realized sensor was connected to the LCR meter (Agilent 4263B) for capacitance measurement. The capacitance measurement was carried out with an ac voltage of 1 V at 1 kHz frequency.

## 4. Conclusions

In conclusion, we followed a straightforward and cost-effective fabrication method to realize a sensitive, stable, and reliable porous PDMS-based pressure sensor functionalized by MWCNTs. The complete electromechanical characterization of the sensor shows a potential application in different fields, including tactile monitoring, Morse code application, and physiological monitoring. In addition, the CPS demonstrates a proximity sensing capacity enabling the sensor’s ability for both contact and non-contact sensing abilities. The high performance of the pressure sensor comes from the synergistic effect of the porous polymer and the MWCNT composite as the dielectric layer. The porosity improves the compressibility of the dielectric layer, and the addition of MWCNTs with the polymer improves the dielectric permittivity of the polymer. To improve the compressibility of the polymer, acetone was used as a polymer diluter, which has an added influence on the compressibility improvement. The pressure sensitivity increases with the addition of the MWCNTs to the solution, and the highest sensitivity could be achieved by adding 1.6% of MWCNTs with the PDMS. Due to the ultrahigh-pressure sensitivity at the low-pressure regime, the sensor has an ultralow detection limit of 1.46 Pa and excellent stability of 9000 cycles, proving the pressure sensor’s reliability for numerous applications. The pressure sensor’s outstanding performance allows for tactile sensing detection and subtle pulse waveform monitoring. Moreover, proximity sensing allows the sensor to monitor object detection at a proximity range of 12 inches. We summarize that the sensor can be employed for physiological monitoring applications and hands-free applications that could improve the fight against diseases.

## Figures and Tables

**Figure 1 molecules-27-06872-f001:**
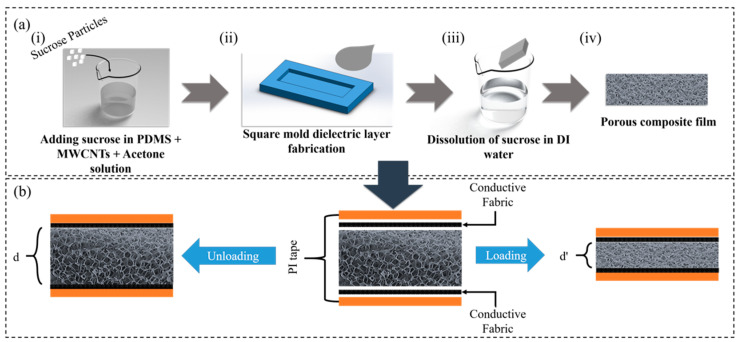
(**a**) Schematic illustration of the fabrication process of porous dielectric layer. (**b**) Pressure-sensing mechanism of the CPS with porous dielectric layer.

**Figure 2 molecules-27-06872-f002:**
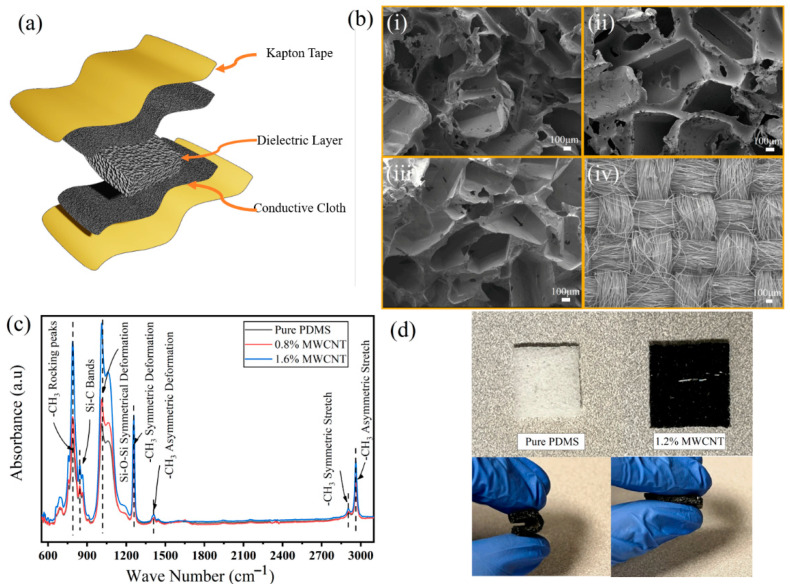
(**a**) Schematic illustration of the sensor. (**b**) SEM images of the porous MWCNT−PDMS structures and conductive textile electrode. (**i**–**iv**) SEM images of pure porous PDMS, 1.2 −MWCNT−PDMS, 2−MWCNT−PDMS, and conductive textile electrode, respectively. (**c**) FTIR spectra of the porous PDMS with different MWCNT concentrations. (**d**) Digital photographs showing pure porous PDMS and 1.2% MWCNT−PDMS composite and the flexibility of the dielectric layer.

**Figure 3 molecules-27-06872-f003:**
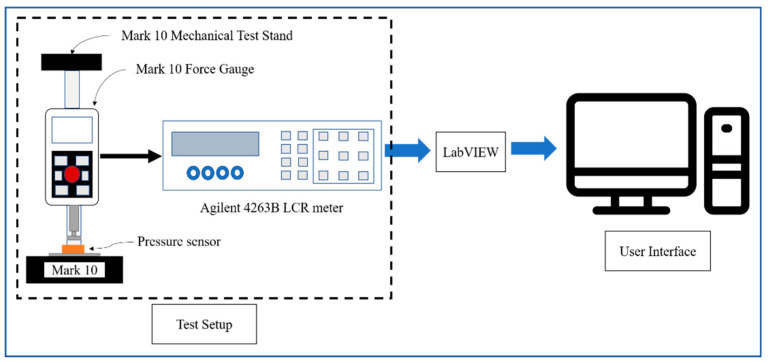
Schematic of the sensor testing set up for electromechanical characterization.

**Figure 4 molecules-27-06872-f004:**
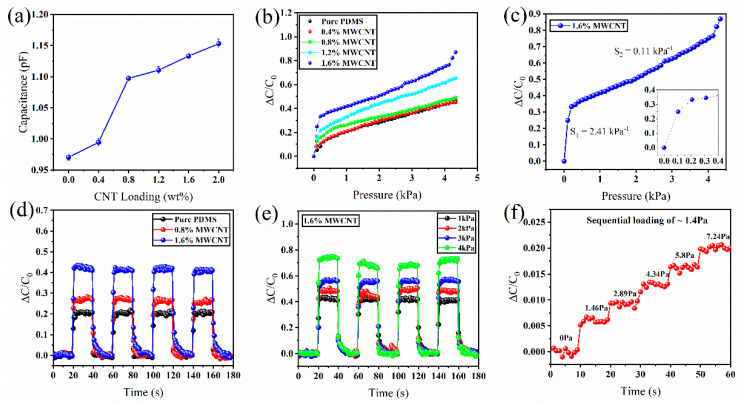
Electromechanical characterization of the MWCNT−PDMS−based pressure sensor. (**a**) Dependence of the initial capacitance based on the concentration of the MWCNTs. (**b**) The relative capacitance change (Δ*C*/*C*_0_) vs. pressure for the MWCNT−PDMS−based sensor with varying concentrations of MWCNT in wt.%. (**c**) The plot shows the relative capacitance change (Δ*C*/*C*_0_) vs. pressure for the MWCNT−PDMS sensor with 1.6 wt.% of MWCNTs. The inset shows the pressure sensitivity at the low−pressure range. (**d**) Loading-unloading profiles for the MWCNT−PDMS−based sensor with varying concentrations of MWCNTs under 1 kPa pressure. (**e**) Loading-unloading profiles for MWCNT−PDMS−based sensor with 1.6 wt.% MWCNT concentration under different pressures. (**f**) Sequential loading of ~1.40 Pa pressure on the pressure sensor showing a minimum detection limit of 1.46 Pa.

**Figure 5 molecules-27-06872-f005:**
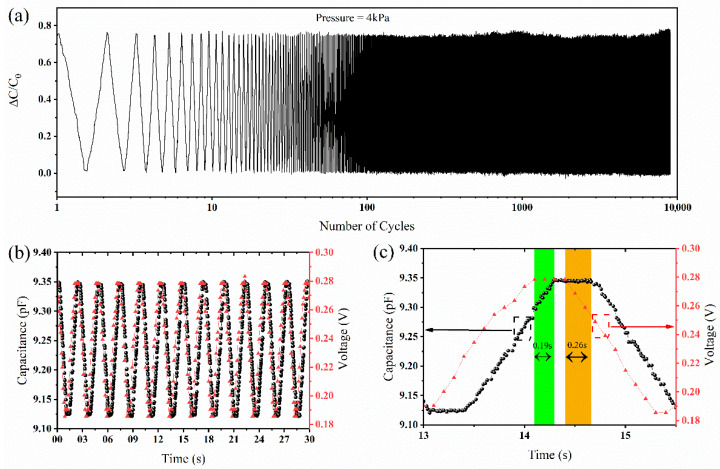
(**a**) Stability test of the MWCNT-PDMS-based pressure sensor showing stable performance for 9000 cycles. (**b**) Comparison of sensor performance between the CPS and a commercial sensor. (**c**) Delay between the response of the commercial sensor and the realized CPS.

**Figure 6 molecules-27-06872-f006:**
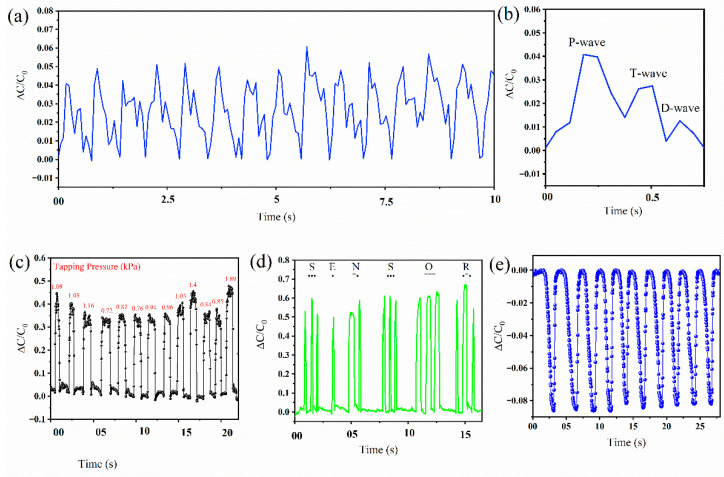
Application of the MWCNT-PDMS-based sensor for real−time pressure monitoring. (**a**) Real-time monitoring of arterial pulse waveforms from the wrist artery. (**b**) Single pulse waveform showing P−wave, T−wave, and D−wave. (**c**) Detection of finger tapping with irregular intervals. (**d**) Generation of Morse code for signal generation with long and short pressing on the sensor. (**e**) The capability of proximity sensing for hands-free applications.

**Table 1 molecules-27-06872-t001:** Comparison among recent studies.

Material	Sensitivity		Year	Reference
	Region-I	Region-II		
Porous PDMS	0.18 kPa^−1^		2021	[13]
Porous PDMS	0.3 kPa^−1^ (<50 Pa)	0.0032 kPa^−1^ (0.2–1 MPa)	2021	[19]
Polyurethane Sponge, GNP	0.062 kPa^−1^ (<1 kPa)	0.033 kPa^−1^ (>1 kPa)	2019	[31]
Porous Ecoflex	0.01 kPa^−1^ (<250 kPa)	0.0009 kPa^−1^ (>250 kPa)	2016	[14]
Porous PDMS	1.18 kPa^−1^ (<20 Pa)		2016	[28]
Porous PDMS	0.86 kPa^−1^		2019	[17]
Porous PDMS, MWCNT	2.41 kPa^−1^ (<50 Pa)	0.11 kPa^−1^ (>1 kPa)	2022	This work

## Data Availability

The raw/processed data required to reproduce the study cannot be shared at this time as the data also form part of an ongoing research.

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
