# Peer review of "Wearable Capacitive Pressure Sensor for Contact and Non-Contact Sensing and Pulse Waveform Monitoring"

_molecules, 2022, doi:10.3390/molecules27206872_

Round 1
Reviewer 1 Report
This is an interesting paper describing the materials fabrication for a PDMS/MWCNT composite dielectric layer and its application in pressure sensors. The paper is well written with a clear presentation style and the data presented show strong application potential of the reported dielectric material and sensor.
Reviewer 2 Report
This manuscript reports a sensor based on porous PDMS and MWCNT composite-based dielectric layer. It demonstrates high sensitivity, low detection limit and low-cost fabrication method. The experimental design of this work is sound, and writing style is also logical. I support the publication of this paper, with minor comments as follows.
1. It is true that the addition of MWCNT can improve the sensitivity of the device. However, there should be a percolation threshold over which the dielectric layer becomes conductive and the device fails. The authors should add more experiments to show the limit value of MWCNT amount and discuss more in the manuscript.
2. More comparison references should be added in Table 1.
3. The grammar and spelling should be carefully checked again.
Reviewer 3 Report
In the manuscript, the authors report on the template-assisted method for the fabrication of the capacitive sensor. Making use of glucose or sucrose particles or even common salt with ordinary polymers and leaching them for the preparation of the sponge-like microstructure and sandwiching it in between conductive electrodes for capacitive pressure sensors is not a novel approach! The authors have to mention the novelty of this work explaining how it differs from previous literature. Overall, the manuscript mainly needs novelty along with revision addressing the following major concerns.
Comments:
- The incorporation of MWCNT increases the elastic modulus of the composite sponge! Sensitivity is not only dependent on the dielectric permittivity but also governed by the modulus ─ the lower the modulus/compressibility, the higher will be the sensitivity (how easily the sponge can be compressed with external force/pressure). On the contrary, the authors have to compromise or trade off between dielectric permittivity and the elastic modulus. In addition, a detailed investigation of the percolation threshold of the MWCNT for dielectric permittivity is also important! A detailed investigation of Figure 4a (also consistent with Figure S1) would be more interesting to the readership.
- How much is the sucrose particle’s size? How does the sucrose particle size and uniformity affect the porosity and hence sensor performance?
- What are the purpose of using different electrode materials-ITO/PET and conductive fabric (Figure 1b) and Kapton tape with conductive fabrics (Figure 2a)? Did the authors fabricate two sets of sensors and reported one?
- The long-term durability of the sensor is also dependent on the externally applied force/pressure, meaning that the stronger the applied pressure, the lower will be the durability. The statement “Besides, the pressure….” in the Abstract does not support the title of the manuscript as it is intended to apply for pulse waveform monitoring, which may not match with “reliability for robust applications”! In addition, as the sensor is applied for as tactile sensing, Morse code generator, etc., which does not limit the TITLE of the work to be specific!
- The statement “However, such delicate…” is not quite following the previous sentence because the preceding sentences are about the microstructures, not the microstructured materials!
- What do the authors mean by “triangular loading profile” in the caption of Figure 5? The reviewer cannot distinguish the capacitive response to the low-and high-pressure frequency in Figures 5b and 5c, respectively! Secondly, the reviewer is wonder why the authors compared the response of their CPS with a commercial piezoresistive sensor and expressed the response into volts! Are the comparison and the data plots trying to superimpose the capacitance and voltage response appropriate?
- Are each term in equations (1) and (2) defined? Please ensure equation (2) is correctly typed!
- What is the mechanism behind the proximity sensing of the sensor? If so, is the real-time application of the sensor disturbed or interfered with by human skin or fingers? Please indicate the mechanism of how CPS is working as proximity sensing.
- The pulse waveform recorded in Figure S3 doesn’t bear any information, even the hard-to-see shape of the pulse!
Reviewer 4 Report
In the manuscript entitled "Wearable Capacitive Pressure Sensor for Pulse Waveforms Monitoring", the authors presented the formation of low-cost pressure sensors with porous elastomer and carbon nanotubes, and demonstrated them for tactile sensing and physiological monitoring. The authors' main contributions include (1) the development of high-sensitivity pressure sensors with low-cost and commercially available materials, and (2) its various demonstration.
However, from my point of view, this manuscript can be published in Molecules, but the revision is required due to the following reasons:
1. It seems that the contents of this manuscript are not matched the scope of the journal. The journal Molecules is mainly based on chemistry. However, it is hard to find the material chemistry in this manuscript, and it was focused on the fabrication of pressure sensing devices and their characterizations.
2. In the introduction section, the authors mainly discussed the necessity of pressure sensing for cardiovascular monitoring. However, there is only data on arterial pulse monitoring which are detectable by conventional and commercial pressure sensors, and this single result seems to fail to present the benefit of this device for cardiovascular monitoring. Authors should add more characterizations and data about cardiovascular monitoring.
3. In Figure 5a, the authors explained that there was a stable performance without degradation during 10,000 cycles. However, Figure 5a shows significant fluctuation in the change in dC/C0. The authors should discuss about the cyclic test results, such as: why the initial dC/C0 level went down, why the pressed dC/C0 level went up, and why the dC/C0 starts at about 0.3, rather than 0?

Round 2
Reviewer 3 Report
The authors have satisfactorily revised the manuscript. Now, it can be considered for publication in this journal.
Reviewer 4 Report
The authors addressed the comments appropriately, and I think the revised manuscript can be published.